# Risk of Seven-Day Worsening and Death: A New Clinically Derived COVID-19 Score

**DOI:** 10.3390/v14030642

**Published:** 2022-03-20

**Authors:** Alessia Cruciata, Lorenzo Volpicelli, Silvia Di Bari, Giancarlo Iaiani, Bruno Cirillo, Francesco Pugliese, Daniela Pellegrino, Gioacchino Galardo, Gloria Taliani

**Affiliations:** 1Infectious and Tropical Medicine Unit, Department of Public Health and Infectious Diseases, “Policlinico Umberto I” Hospital, Sapienza University of Rome, Viale del Policlinico 155, 00161 Rome, Italy; a.cruciata@policlinicoumberto1.it (A.C.); lorenzo.volpicelli@uniroma1.it (L.V.); silvia1724.dibari@gmail.com (S.D.B.); giancarlo.iaiani@uniroma1.it (G.I.); 2Department of Surgery “Pietro Valdoni”, “Policlinico Umberto I” Hospital, Sapienza University of Rome, Viale del Policlinico 155, 00161 Rome, Italy; b.cirollo@policlinicoumberto1.it; 3Department of General Surgery and Surgical Specialities “Paride Stefanini”, “Policlinico Umberto I” Hospital, Sapienza University of Rome, Viale del Policlinico 155, 00161 Rome, Italy; f.pugliese@policlinicoumberto1.it; 4Pulmonology, Respiratory and Critical Care Unit, Department of Public Health and Infectious Diseases, “Policlinico Umberto I” Hospital, Sapienza University of Rome, Viale del Policlinico 155, 00161 Rome, Italy; d.pellegrino@policlinicoumberto1.it; 5Medical Emergency Unit, Department of Emergency, “Policlinico Umberto I” Hospital, Sapienza University of Rome, Viale del Policlinico 155, 00161 Rome, Italy; g.galardo@policlinicoumberto1.it

**Keywords:** SARS-CoV-2, COVID-19, risk of death, risk of worsening, clinical score, prognostic score

## Abstract

This monocentric, retrospective, two-stage observational study aimed to recognize the risk factors for a poor outcome in patients hospitalized with SARS-CoV-2 infection, and to develop and validate a risk score that identifies subjects at risk of worsening, death, or both. The data of patients with SARS-CoV-2 infection during the first wave of the pandemic were collected and analyzed as a derivation cohort. Variables with predictive properties were used to construct a prognostic score, which was tried out on a validation cohort enrolled during the second wave. The derivation cohort included 494 patients; the median age was 62 and the overall fatality rate was 22.3%. In a multivariable analysis, age, oxygen saturation, neutrophil-to-lymphocyte ratio, C-reactive protein and lactate dehydrogenase were independent predictors of death and composed the score. A cutoff value of 3 demonstrated a sensitivity (Se), specificity (Sp), positive predictive value (PPV) and negative predictive value (NPV) of 93.5%, 68.5%, 47.4% and 97.2% for death, and 84.9%, 84.5%, 79.6% and 87.9% for worsening, respectively. The validation cohort included 415 subjects. The score application showed a Se, Sp, PPV and NPV of 93.4%, 61.6%, 29.5% and 98.1% for death, and 81%, 76.3%, 72.1% and 84.1% for worsening, respectively. We propose a new clinical, easy and reliable score to predict the outcome in hospitalized SARS-CoV-2 patients.

## 1. Introduction

SARS-CoV-2, a new enveloped RNA virus, has been plaguing the world since December 2019 with the pandemic disease COVID-19, which is characterized by a broad clinical spectrum and resulted in about 5.7 million deaths by February 2022 [1]. Overall, 90% of infections are uncomplicated and do not require hospitalization [2]. On the other side, meta-analyses reported, among hospitalized patients, a pooled rate of intensive care unit (ICU) admission ranging from 10.9% to 26% [3,4] and a 31% prevalence of ICU mortality [4]. Many countries are still struggling with a number of cases currently overburdening health systems [5]. Moreover, while only 10% of people in low-income countries have received at least one dose of a SARS-CoV-2 vaccination [6], new variants of concern, capable of cracking vaccine defense, await in ambush [7] and periodically trigger new pandemic waves [8]. Although vaccine distribution, childhood paucisymptomatic infection and herd immunity will progressively reduce virus transmission and disease impact, it is largely presumed that “SARS-CoV-2 is here to stay”, becoming in the next few years an endemic disease with occasional flare-ups [9].

To help the global sanitary system in handling actual pandemic and future outbreaks, it is necessary to correctly assign the available resources, that is, to promptly distinguish subjects demanding critical care unit beds from those requiring common ward or even home care. Since the onset of the pandemic, many scores have been developed with the purpose of forecasting the severity and outcome of COVID-19 syndrome. The last update of the BMJ living systematic review stated that almost all models that have been proposed remain at a high or unclear risk of bias [10]; thus, research in this field is still of importance. Here, we propose a new, reliable and simple clinical score to predict the risk of worsening and death in hospitalized COVID-19 patients.

## 2. Materials and Methods

### 2.1. Study Endpoints and Design

The aims of the study were (1) to identify the risk factors for any-cause mortality and the clinical deterioration in patients hospitalized with SARS-CoV-2 infection; (2) to develop a risk score able to easily identify patients at risk of a worsening or poor outcome; and (3) to validate this score. We performed a retrospective, single-centre, two-stage study on patients with SARS-CoV-2 infection hospitalized in the urban teaching hospital, Policlinico Umberto I of Rome.

The data of subjects admitted during the first wave of the pandemic in Italy, 1 March to 30 June 2020, were retrospectively collected as a derivation cohort. Then, patients hospitalized between 1 October and 30 November 2020, a time period corresponding to the first phase of the second wave of the pandemic in Italy, were retrospectively considered as a validation cohort. The data sources were clinical charts and hospital electronic records. Each record was checked independently by two researchers. Only anonymized data were collected and managed. The study was approved by the local Ethics Committee, which granted a waiver to the informed consent of patients because of safety risk and the retrospective nature of the study.

### 2.2. Participants, Variables and Outcome

All adults (≥18 years) with laboratory-confirmed SARS-CoV-2 infection (nasopharyngeal swab or bronchoalveolar lavage), hospitalized in Policlinico Umberto I of Rome and with available demographic, anamnestic, clinical and biochemical data recorded at the entrance to the emergency department were included. The exclusion criteria were the lack of a positive sample for SARS-CoV-2, death that occurred before the first clinical and biochemical assessment in the emergency department and a lack of data concerning the final outcome.

The data collected included: demographic (gender, age), comorbidity (COPD, hypertension, cardiovascular and/or cerebrovascular disease, diabetes, chronic kidney disease (CKD), chronic liver disease, obesity, immunodeficiency, active malignancy), presenting history (fever, conjunctivitis, cough, headache, dyspnea, asthenia, confusion, ageusia/anosmia, number of days from onset of symptoms to hospitalization) and laboratory parameters (blood oxygen saturation measured on pulse-oximeter while breathing room air, complete blood count, liver and kidney function, coagulation, inflammatory biomarkers). Patients were considered to be affected by cardiovascular disease when a previous diagnosis of ischaemic or non-ischaemic cardiac disorder, cerebrovascular disease or peripheral arterial disease was reported in the clinical record.

Each cohort (derivation and validation) was analyzed in two different ways. Firstly, the patients were grouped according to the final outcome of the hospitalization (discharge vs. death). Secondly, the same cohort was studied according to the clinical evolution at a reassessment performed after seven days of hospitalization (stability vs. worsening within seven days from hospitalization). All patients that deceased before the seventh day of hospitalization were considered as worsened in this latter analysis. Patients received supportive and antiviral treatment according to the national and local protocols available at the time [11].

Worsening was defined as:Escalating need of oxygen delivery: switch from no need to a need of oxygen; switch from a simple nasal cannula or Venturi face mask to a high-flow nasal cannula or helmet noninvasive ventilation; switch from noninvasive ventilation to invasive ventilation;Transfer to a ward with a higher level of assistance: from the regular ward to semi-intensive or intensive care units; from the semi-intensive to the intensive care unit;Occurrence of a severe complication (diagnosed and treated during hospitalization according to local guidelines): acute pulmonary embolism; sepsis; need for vasopressors support; acute coronary syndrome; ischaemic or hemorrhagic stroke; acute kidney injury requiring renal replacement therapy.

### 2.3. Statistical Analysis

Categorical variables were expressed as counts and percentages and compared through a Chi-squared test or Fisher exact test. Continuous variables were described as median and interquartile range (IQR). A Kolmogorov–Smirnov test was used to check for normality. A Student t test and Mann–Whitney U test confronted normally distributed or skewed continuous variables, respectively. Univariable and multivariable analyses were performed to identify the risk factors for mortality. In order to build a risk score, we used a regression coefficient-based scoring system as previously described [12]: each variable that maintained significance in the multivariable analysis was assigned a point value corresponding to the beta-coefficient, rounded to the nearest integer, to derive weights.

The discriminative power of the derived score was assessed with the area under the receiver operating characteristic (ROC) curve (AUROC). The optimal cutoff and the corresponding sensitivity (Se), specificity (Sp), positive and negative predictive values (PPV and NPV) were assigned using the Youden test. The statistical analysis was performed using SPSS software version 25 (IBM, Armonk, NY, USA).

## 3. Results

In total, 512 adult patients accessed the emergency department of Policlinico Umberto I from 1 March to 30 June 2020, were diagnosed with COVID-19 and were hospitalized. Eighteen were excluded for the study according to the exclusion criteria (unknown outcome as they were transferred to a different hospital), so that, finally, we analyzed data from 494 individuals. In twelve of these, we could not find information on their clinical evolution seven days after their hospitalization, so that they were not analyzed for worsening risk factors, but only for the final outcome. Baseline characteristics of the entire population, differential features according to the seven-day re-evaluation and final outcomes are reported in Table 1.

### 3.1. Overall Cohort

The median age of our cohort was 62 years (Table 1). Grouped by age, 137 patients (27.7% of the entire population) were aged <50 years, 132 (26.7%) were 50–64 years, 89 (18%) were 65–74 years, 85 (17.2%) were 75–84 years and 51 (10.3%) were over 85 years. Male subjects were 267 (54%); fever (76.3%), cough (50.4%), dyspnea (44.6%) and asthenia (22.6%) were the most frequently reported symptoms at entrance. Hypertension, vascular disease, COPD and diabetes were the most common pre-existing comorbidities. A mild-to-moderate increase in inflammatory biomarkers on laboratory assessment, including CRP, LDH and D-dimer, was very common.

### 3.2. Re-Evaluation at the Seventh Day: Stable/Improved vs. Worsened/Dead

Seven-day re-evaluations were available for 482 (98%) individuals in the derivation cohort, 201 (41.7%) of whom had progressed toward deterioration of their clinical conditions, or even death, within seven days from hospitalization. Many significant differences in the baseline characteristics between stable/improved and worsened/dead patients were found (Table 1). Those individuals whose condition worsened or who died in the first seven days of hospitalization were older, more frequently male, febrile, and had higher respiratory and heart rates and lower oxygen saturation in room air at their admission to the emergency department. As for symptoms, fever and dyspnea were more frequent in the group of worsened/dead, while headache and ageusia/anosmia were more frequent among the stable/improved patients. All the considered comorbidities, except for immunodeficiency and obesity, were significantly more represented in the worsened/dead group. Laboratory evaluations demonstrated a more severe alteration in the total and differential leukocyte counts in worsened/dead subjects, mainly characterized by higher white blood counts, higher neutrophil/lymphocyte ratios and lower haemoglobin. They also had significantly higher levels of inflammatory biomarkers and lower albumin.

### 3.3. Final Outcome: Survivors vs. Deceased Patients

Overall, the in-hospital fatality rate was 22.3% (110 out of 494 patients) in the derivation cohort. The mortality rates among patients <50, 50–64, 64–74, 75–84 and ≥85 years were 2.9%, 13.6%, 34.8%, 36.5% and 51%, respectively. At the time of admission, patients who would later die during their hospitalization were significantly older, had higher heart and respiratory rates, were more frequently dyspnoic and had reduced arterial blood saturation in room air. Except for dyspnea, no other COVID-19 symptom frequency differed between the two groups, while none of the patients in the deceased group reported ageusia or anosmia. Again, pre-existing chronic comorbidities were more represented, and blood-count and biochemical alterations were more pronounced, in the group with the worst prognosis (Table 1).

### 3.4. Multivariable Analysis and Factors Independently Associated with Death

All the variables that were found significantly associated (*p* < 0.05) with death in the univariable analysis were included in a multivariable analysis. In the multivariable analysis (Table 2), age ≥ 60 years, room air oxygen saturation ≤ 94%, neutrophils/lymphocytes ratio ≥ 10, C-reactive protein ≥ 10 mg/dL and lactate dehydrogenase ≥ 330 IU/L remained independently associated with negative outcomes.

Specifically, among the variables that were collected at emergency department admission and analyzed, and that resulted in significantly different outcomes, an age ≥ 60 years and an oxygen saturation in room air ≤ 94% were the factors with the greatest impact on survival, being characterized by ORs of 5.043 (CI 2.473–10.284) and 4.555 (2.599–7.982), respectively. Furthermore, a neutrophils/lymphocytes ratio ≥ 10, a C-reactive protein ≥ 10 mg/dL and a lactate dehydrogenase ≥ 330 IU/L also remained independently associated with the outcome of death, showing ORs of 3.023 (1.619–5645), 2389 (1305–4374) and 1861 (1052–3292), respectively.

### 3.5. Score Development

By assigning to each factor a point value that corresponds to the beta-coefficient rounded to the nearest integer, we derived a clinical score that predicts death among inpatients with SARS-CoV-2 infection (Table 2). An ROC analysis (Figure 1A) showed an AUC of 0.879 (95% CI 0.846–0.912), and a cutoff value of 3 demonstrated a sensitivity (Se), specificity (Sp), positive predictive value (PPV) and negative predictive value (NPV) of 93.5%, 68.5%, 47.4% and 97.2%, respectively. No significant differences in terms of final outcome emerged when comparing patients receiving 1 point of the score with those receiving 2 points (*p* = 0.189).

The newly derived score was also applied to the seven-day re-evaluation status in order to test its prediction power for clinical worsening within seven days from hospitalization. The AUC was 0.898 (95% CI 0.868–0.927) (Figure 1C), and the Se, Sp, PPV and NPV were 84.9%, 84.5%, 79.6% and 87.9%, respectively.

### 3.6. Score Validation

The validation cohort was composed of 415 patients admitted because of COVID-19 between 1 October and 30 November 2020. Worsening and final outcome data were available for the entire group. The demographic, anamnestic and clinical features were close to those of the derivation cohort and are reported in Table 3. The worsening rate was similar to the derivation cohort (43.1% vs. 41.7%), while the occurrence of death was lower (14.7% vs. 22.3%). The score was then applied to the validation cohort (Figure 1B,D): the AUC for the outcome of death was 0.830 (95% CI 0.785–0.875), with a Se 93.4%, Sp 61.6%, PPV 29.5% and NPV 98.1%; the AUC for the outcome of worsening was 0.826 (95% CI 0.784–0.868), with a Se 81%, Sp 76.3%, PPV 72.1% and NPV 84.1%.

## 4. Discussion

We described the characteristics of a cohort of SARS-CoV-2 infected patients who were hospitalized during the first period of the pandemic in an Italian teaching hospital, and identified the relevant risk factors for death and for clinical deterioration within seven days from hospitalization. A score with easy-to-obtain clinical characteristics was derived and validated on a second cohort.

At the beginning of the pandemic, patients were hospitalized according to their clinical state at the time of entrance in the emergency department. This way, many people initially allocated to low-intensive care units subsequently had to be transferred to semi-intensive and intensive care units. Moreover, seriously ill patients in peripheral hospitals frequently had to be transferred to central hospitals. Nowadays, many factors are known to contribute to the increased risk for a severe course of the disease, even for a subject with apparently good clinical conditions. An appropriate combination of these factors may represent a tool for the early recognition of individual risk and the effective selection of the level of care commensurate with the potential clinical request. Many clinical scores have been proposed to predict either mortality or disease worsening. As stated, in the last update of the BMJ living systematic review, all models proposed until now remain at a high or unclear risk of bias [10]. Below, we briefly review some of the most relevant scores that were applied in Italy.

The CALL score was developed based on 208 patients admitted to two Chinese hospitals between January and February 2020 to predict disease progression; it ranges from 4 to 13 points, and individuals with severe symptoms at entrance were excluded during enrollment. Similar to our score, it is characterized by a high negative predictive value [13], but unfortunately, its performance when applied to an Italian cohort was not satisfactory [14]. The COMPASS-COVID-19 score was derived based on a cohort of 310 subjects and validated on 120 subjects prospectively enrolled; it is applicable only to patients undergoing LMWH thromboprophylaxis and predicts disease worsening with a Se 94% and Sp 58%. Ranging between 0 and 54 points, it requires an online calculator [15]. The PREDI-CO score was developed and validated on a cohort of hospitalized SARS-CoV-2 infected subjects from eleven hospitals, during the first wave of the pandemic in Italy; it was highly discriminant to predict the risk of subsequent severe respiratory failure, but not of mortality [16]. Fumagalli et al. proposed the COVID-19MRS score for in-hospital mortality based on the data collected in two tertiary hospitals in Northern and Central Italy, but no validation is available [17]. The Brescia-COVID respiratory severity scale is a clinical and radiological dynamic algorithm developed during the first wave of the pandemic in Italy [18] that requires periodic reassessment, with a frequency rate based on subjective clinical judgment. Its predictive power was later validated, demonstrating a good performance in predicting ICU admission [19] and mortality [20].

More recently developed and validated scores were based on larger and often multicentre populations [21,22,23,24]. The specifically developed scores, ISARIC-4C, COVID-GRAM and qCSI, were tested on an Italian cohort of patients aged 60 or older, but did not perform significantly better in predicting death when compared to the standard, longtime validated NEWS [25].

The five variables that we finally assembled in the score represent well-established predictive markers of SARS-CoV-2 infection severity and evolution. In our cohort, age is confirmed to constitute the strongest predictor of COVID-19 prognosis [26]. In fact, when grouped according to age, the mortality rate dramatically increased up to 51% in people older than 85. This result resembles data from other Italian casuistic studies [17] and is in line with a recent analysis of data from 45 countries, confirming a higher infection fatality rate in countries, such as Italy and Japan, with an older demographic [27]. A reduced oxygen saturation (≤94% in room air) defines severe COVID-19 in international guidelines [28]. The neutrophil-to-lymphocyte ratio is recognized as a simply accessible and strong indicator of an imbalance in the immune response and the risk of severity and mortality, although an optimal cutoff is yet to be established [29]. Similarly, CRP and LDH levels account for the intensity of systemic inflammation and tissue damage, respectively, and are known to correlate with critical illness and death in COVID-19 patients [30,31].

Although our score was developed using variables found associated with death, it also performed well when applied to the risk of worsening within seven days from hospitalization. The reduced mortality in the validation cohort probably reflects an improvement in the management and treatment protocols for COVID-19 patients at different timepoints of the pandemic. The characteristics of the proposed score give it broad applicability. In the setting of a local outbreak, a high sensitivity would allow the easy identification of all subjects with an increased risk of severe course and worse prognosis. Differently, in the setting of a pandemic with limited resource availability, a high NPV allows the secure discharge or admittance to low-intensity units of patients with a low risk of death, lightening the burden of the health system. A combined application with a score with different characteristics may increase the yield. Furthermore, the score we are illustrating has the quality of simplicity: it is independent of the radiologic picture, and the clinician only needs a pulse oximeter and routine blood exams to apply this score.

In the present study, the univariable analysis identified some interesting features associated with the prognosis of patients affected by COVID-19, although these were not confirmed in the multivariable analysis. Ageusia and anosmia were significantly more represented in patients with a favorable outcome. This is in line with a subanalysis of the HOPE COVID-19 registry, where olfactory and gustatory dysfunctions were inversely correlated with death [32]. On the contrary, the serum albumin level at hospital entrance was lower in the non-survival group, probably reflecting either abnormal liver function, proteinuria [33], poor nutritional status [34] or even increased thrombotic risk [35]. Differences in immunodeficiency and obesity were probably not found because of low numbers. In agreement with other findings [36], male gender was a risk factor for worsening within seven days from hospitalization. This difference was not significant for the final outcome, probably because of low statistical power. A small group of pregnant women were present both in derivation and validation cohorts, and a favorable outcome was generally observed.

This study has many limitations. As a single-centre, retrospective study, the results may have poor external validity, and sample numerosity is lower than that used for the derivation of other scores. Since the data were collected during the spring and autumn of 2020, these results correspond to the first two SARS-CoV-2 variants of concern that spread in Italy (presumably the wild B1 strain and the B.1.1.7). We decided to exclude smoking status because of poor availability in both the derivation and validation cohorts. Other variables, especially comorbidities and presenting symptoms, could have been missed because of the rapid collection of information usually performed in an emergency department. Finally, it was not possible to analyze the impact of the COVID-19 treatment strategies, which underwent profound changes during the course of the pandemic. Anyway, it can reasonably be assumed that treatments available during the periods considered were mainly supportive.

As strengths of the study, it should be considered that the variables included in the score are operator-independent. The radiologic evaluation was not included in the analysis, as it is not always readily available, but it may integrate the prognostic information provided by the score. Lastly, the score computation is simple and does not require an online calculator.

## 5. Conclusions

We proposed a new clinical, easy-to-use score to predict the early clinical deterioration and final outcome in SARS-CoV-2 infected inpatients. Its high sensitivity and very high negative predictive value conferred good clinical applicability in the cohorts examined here. Further validation is required in the context of the current scenario of the codominance of the Delta (B.1.617.2) and Omicron (B.1.1.529) variants of concern and of the preventive and therapeutic tools available.

## Figures and Tables

**Figure 1 viruses-14-00642-f001:**
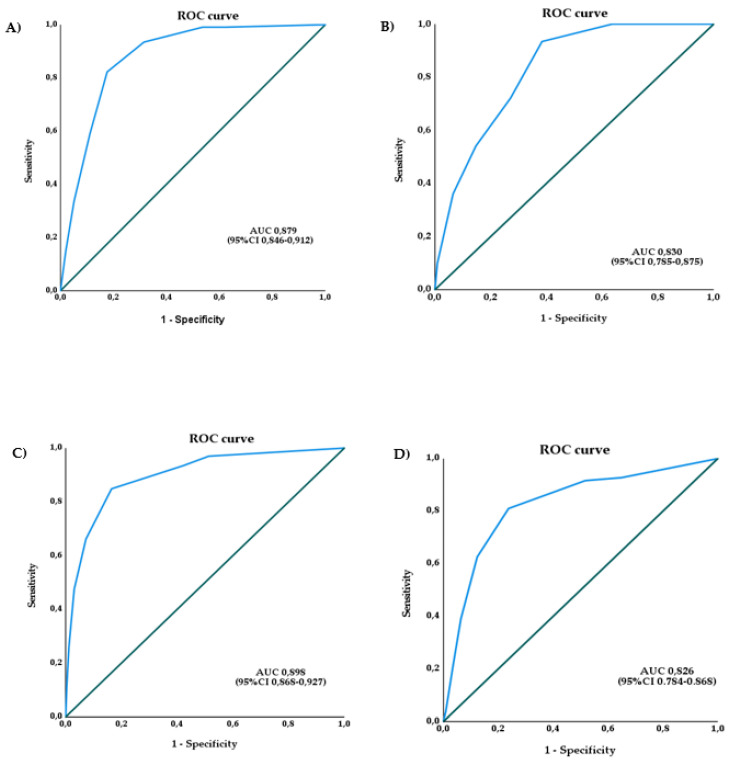
ROC curve for death and for worsening in seven days in the derivation and validation cohorts: (**A**) ROC curve for death in the derivation cohort; (**B**) ROC curve for death in the validation cohort; (**C**) ROC curve for worsening within seven days in the derivation cohort; (**D**) ROC curve for worsening within seven days in the validation cohort. AUC: Area Under the Curve.

**Table 1 viruses-14-00642-t001:** Clinical and laboratory characteristics of the derivation cohort according to seven-day re-evaluation and outcome.

Variable	Overall Cohort (n = 494)	Stable (n = 281)	Worsened (n = 201)	*p* ^a^	Survived (n = 384)	Deceased (n = 110)	*p* ^b^
**Age (years), median (IQR)**	62 (49–76)	54 (39–68)	72 (63–82)	<0.001 *	57 (44–72)	76 (67–84)	<0.001 *
**Gender (male), n (%)**	267 (54.0)	135 (48.0)	124 (61.7)	0.003 *	202 (52.6)	65 (59.1)	0.235
**Days from onset of symptoms, median (IQR)**	6 (3–10)	6 (3–9)	6 (3–10)	0.463	6.5 (3.0–10.0)	6.0 (3.0–8.5)	0.185
**Body temperature °C, median (IQR)**	37.0 (36.2–37.8)	36.8 (36.1–37.5)	37.4 (36.4–38.0)	<0.001 *	37.0 (36.2–37.7)	37.4 (36.4–38.0)	0.014 *
**Respiratory rate, median (IQR)**	18 (16–22)	18 (16–18)	20 (18–25)	<0.001 *	18 (16–20)	22 (18–26)	<0.001 *
**Heart rate, median (IQR)**	89 (80–100)	86 (78–100)	90 (80–104)	0.026 *	87 (79–100)	90 (80–109)	0.033 *
**Oxygen saturation in room air, median (IQR)**	97 (94–98)	98 (96–99)	93 (88–97)	<0.001 *	97 (95–99)	92 (86–95)	<0.001 *
**Fever, n (%)**	358 (76.3)	197 (71.9)	160 (82.5)	0.008 *	279 (75.8)	79 (78.2)	0.692
**Conjunctivitis, n (%)**	11 (2.4)	8 (2.9)	3 (1.6)	0.538	7 (1.9)	4 (4.0)	0.260
**Headache, n (%)**	50 (10.7)	37 (13.6)	13 (6.7)	0.022 *	39 (10.7)	11 (11.0)	1.0
**Cough, n (%)**	236 (50.4)	139 (50.7)	97 (50.3)	0.925	184 (50.0)	52 (52.0)	0.736
**Asthenia, n (%)**	106 (22.6)	67 (24.5)	38 (19.7)	0.261	81 (22.0)	25 (25.0)	0.590
**Diarrhea, n (%)**	70 (15.0)	40 (14.7)	30 (15.5)	0.794	58 (15.8)	12 (12.0)	0.430
**Dyspnea, n (%)**	209 (44.6)	91 (33.2)	118 (60.8)	<0.001 *	138 (37.5)	71 (70.3)	<0.001 *
**Confusion, n (%)**	34 (7.3)	19 (7.0)	15 (7.8)	0.857	22 (6.0)	12 (12.0)	0.051
**Ageusia/Anosmia, n (%)**	37 (7.9)	35 (12.8)	1 (0.5)	<0.001 *	37 (10.1)	0	<0.001 *
**Chronic obstructive pulmonary disease, n (%)**	89 (19.1)	39 (14.4)	50 (25.8)	0.003*	55 (15.2)	34 (33.3)	<0.001 *
**Diabetes, n (%)**	89 (19.1)	33 (12.2)	56 (28.9)	<0.001 *	56 (15.4)	33 (32.4)	<0.001 *
**Hypertension, n (%)**	208 (44.6)	84 (31.0)	124 (63.9)	<0.001 *	143 (39.3)	65 (63.7)	<0.001 *
**Liver disease, n (%)**	20 (4.1)	6 (2.1)	14 (7.0)	0.016 *	12 (3.1)	8 (7.3)	0.177
**Cardiovascular disease, n (%)**	157 (33.9)	73 (26.9)	84 (44.0)	<0.001 *	95 (26.2)	62 (61.4)	<0.001 *
**Active malignancy, n (%)**	77 (16.6)	36 (13.3)	41 (21.1)	0.031 *	51 (14.0)	26 (25.5)	0.010 *
**Chronic kidney disease, n (%)**	28 (6.0)	8 (3.0)	20 (10.3)	0.001 *	13 (3.6)	15 (14.7)	<0.001 *
**Immunodeficiency, n (%)**	25 (5.4)	12 (4.4)	13 (6.7)	0.304	14 (3.9)	11 (10.8)	0.011 *
**Obesity, n (%)**	42 (9.0)	22 (8.1)	20 (10.3)	0.512	32 (8.8)	10 (9.8)	0.701
**Pregnancy, n (%)**	7 (1.4)	7 (2.5)	0 (0)	0.045 *	7 (1.8)	0 (0)	0.355
**White blood cell (×10^9^/L), median (IQR)**	6.1 (4.4–8.8)	5.8 (4.2–7.7)	6.8 (4.9–10.5)	<0.001 *	5.8 (4.3–7.8)	8.3 (5.1–12.0)	<0.001 *
**Lymphocytes (×10^9^/L), median (IQR)**	1.0 (0.6–1.5)	1.2 (0.8–1.7)	0.7 (0.5–1.0)	<0.001 *	1.1 (0.7–1.6)	0.7 (0.4–1.0)	<0.001 *
**Neutrophils (×10^9^/L), median (IQR)**	4.3 (3.0–6.6)	3.8 (2.7–5.4)	5.4 (3.4–8.8)	<0.001 *	4.0 (2.8–5.8)	6.2 (3.8–10.5)	<0.001 *
**Neutrophils/lymphocytes ratio, median (IQR)**	4.3 (2.5–8.0)	3.1 (2.1–5.0)	7.5 (4.3–13.6)	<0.001 *	3.8 (2.2–6.4)	9.0 (4.6–17.9)	<0.001 *
**Platelets (×10^9^/L), median (IQR)**	201 (161–263)	210 (166–263)	193 (152–248)	0.058	202 (164–261)	196 (144–268)	0.348
**Haemoglobin (g/dL), median (IQR)**	13.6 (11.9–14.7)	13.9 (12.5–14.9)	13.2 (11.1–14.3)	<0.001 *	13.8 (12.4–14.9)	11.9 (10.2–13.7)	<0.001 *
**Serum creatinine (mg/dL), median (IQR)**	0.90 (0.7–1.1)	0.8 (0.7–1.0)	1.0 (0.8–1.3)	<0.001 *	0.9 (0.7–1.0)	1.1 (0.8–1.5)	<0.001 *
**C-reactive protein (mg/dL), median (IQR)**	2.7 (0.7–8.4)	1.2 (0.2–3.5)	7.8 (2.8–15.8)	<0.001 *	1.6 (0.4–6.0)	10.2 (3.7–19.9)	<0.001 *
**Lactate dehydrogenase (IU/L), median (IQR)**	286 (216–387)	233 (195–299)	373 (289–493)	<0.001 *	268 (208–341)	389 (274–521)	<0.001 *
**Albumin (g/dL), median (IQR)**	3.7 (3.2–4.1)	3.9 (3.6–4.2)	3.4 (2.9–3.7)	<0.001 *	3.8 (3.5–4.2)	3.1 (2.8–3.6)	<0.001 *
**International normalized ratio, median (IQR)**	1.03 (0.98–1.09)	1.02 (0.98–1.07)	1.04 (0.98–1.12)	0.007 *	1.02 (0.98–1.07)	1.06 (0.99–1.16)	<0.001 *
**D-dimer (µg/L), median (IQR)**	848 (440–1967)	551 (360–1133)	1478 (760–3760)	<0.001 *	647 (398–1285)	2244 (1265–4382)	<0.001 *
**Aspartate aminotransferase (IU/L), median (IQR)**	26 (19–40)	23 (18–31)	33 (24–53)	<0.001 *	25 (19–37)	30 (21–55)	0.002 *

Abbreviations: IQR: interquartile range. ^a^ univariable analysis for stable vs. worsened patients; ^b^ univariable analysis for survived vs. deceased patients; * statistically significant.

**Table 2 viruses-14-00642-t002:** Multivariable analysis and score development.

Scoring Factor	Odd Ratio	95% CI	*p* Value	*β*-Coefficient	Score Assigned *
**Age ≥ 60 years**	5043	2473–10,284	<0.001	1618	**2**
**SaO2 ≤ 94%**	4555	2599–7982	<0.001	1516	**2**
**N/L ≥ 10**	3023	1619–5645	<0.001	1106	**1**
**CRP ≥ 10 mg/dL**	2389	1305–4374	0.005	0.871	**1**
**LDH ≥ 330 IU/L**	1861	1052–3292	0.033	0.621	**1**

Abbreviations: CRP: C-reactive protein; LDH: lactate dehydrogenase; N/L: neutrophils-to-lymphocytes ratio; SaO2: arterial blood oxygen saturation in room air. * For each patient, this numerical value is assigned if the scoring factor is present; otherwise, zero is assigned. The individual values are then added up, generating the overall value of the score.

**Table 3 viruses-14-00642-t003:** Clinical and laboratory characteristics of the validation cohort according to seven-day re-evaluation and outcome.

Variable	Overall Cohort (n = 415)	Stable (n = 236)	Worsened (n = 179)	Survived (n = 354)	Deceased (n = 61)
**Age (years), median (IQR)**	63 (52–75)	59 (47–70)	70 (58–78)	60 (51–72)	79 (71–86)
**Gender (male), n (%)**	259 (62.4)	144 (61.0)	115 (64.2)	222 (62.7)	37 (60.7)
**Days from onset of symptoms, median (IQR)**	6 (3–8)	5.5 (4.0–8.0)	6.0 (3.0–8.0)	6.0 (4.0–8.0)	4.0 (2.0–7.0)
**Oxygen saturation in room air, median (IQR)**	96 (94–98)	97 (96–98)	94 (91–96)	97 (94–98)	93 (89–95)
**Fever, n (%)**	319 (76.9)	171 (72.5)	148 (82.7)	274 (77.4)	45 (73.8)
**Conjunctivitis, n (%)**	22 (5.3)	8 (3.4)	14 (7.8)	16 (4.5)	6 (9.8)
**Headache, n (%)**	92 (22.2)	58 (24.6)	34 (19.0)	80 (22.6)	12 (19.7)
**Cough, n (%)**	202 (48.7)	117 (49.6)	85 (47.5)	179 (50.6)	23 (37.7)
**Asthenia, n (%)**	137 (33.0)	73 (30.9)	64 (35.8)	112 (31.6)	25 (41.0)
**Diarrhea, n (%)**	65 (15.7)	36 (15.3)	29 (16.2)	60 (16.9)	5 (8.2)
**Dyspnea, n (%)**	220 (53.0)	87 (36.9)	133 (74.3)	172 (48.6)	48 (78.7)
**Confusion, n (%)**	40 (9.6)	22 (9.3)	18 (10.1)	23 (6.5)	18 (29.5)
**Ageusia/Anosmia, n (%)**	29 (7.0)	19 (8.1)	10 (5.6)	27 (7.6)	2 (3.3)
**Chronic obstructive pulmonary disease, n (%)**	70 (16.9)	33 (14.0)	37 (20.7)	51 (14.4)	19 (31.1)
**Diabetes, n (%)**	76 (18.3)	41 (17.4)	35 (19.6)	59 (16.7)	17 (27.9)
**Hypertension, n (%)**	206 (49.6)	95 (40.3)	111 (62.0)	165 (46.6)	41 (67.2)
**Liver disease, n (%)**	15 (3.6)	12 (5.1)	3 (1.7)	12 (3.4)	3
**Cardiovascular disease, n (%)**	152 (36.6)	82 (34.7)	70 (39.1)	118 (33.3)	34 (55.7)
**Active malignancy, n (%)**	53 (12.7)	28 (11.9)	25 (14.0)	39 (11.0)	14 (22.9)
**Chronic kidney disease, n (%)**	29 (7.0)	11 (4.7)	18 (10.1)	15 (4.2)	14 (23.0)
**Immunodeficiency, n (%)**	38 (9.2)	21 (8.9)	17 (9.5)	32 (9.0)	6 (9.8)
**Obesity, n (%)**	23 (5.5)	9 (3.8)	14 (7.8)	19 (5.4)	4 (6.6)
**Pregnancy, n (%)**	9 (2.2)	8 (3.4)	1 (0.6)	9 (2.5)	0 (0)
**White blood cell (×10^9^/L), median (IQR)**	6.9 (4.9–9.0)	6.6 (4.8–8.5)	7.2 (5.5–10.3)	6.7 (5.0–8.6)	8.9 (6.1–13.1)
**Lymphocytes (×10^9^/L), median (IQR)**	0.9 (0.7–1.3)	1.1 (0.7–1.6)	0.8 (0.6–1.1)	1.0 (0.7–1.3)	0.7 (0.6–1.1)
**Neutrophils (×10^9^/L), median (IQR)**	5.1 (3.5–7.3)	4.7 (3.2–6.3)	5.8 (4.1–8.9)	4.8 (3.4–6.6)	7.7 (4.3–11.7)
**Neutrophils/lymphocytes ratio, median (IQR)**	5.1 (3.1–9.4)	4.1 (2.6–6.5)	7.8 (4.4–13.9)	5.0 (3.1–8.1)	10.4 (3.7–17.6)
**Platelets (×10^9^/L), median (IQR)**	212 (164–287)	219 (172–292)	195 (153–274)	213 (166–290)	194 (145–260)
**Haemoglobin (g/dL), median (IQR)**	13.8 (12.5–15.0)	14.0 (12.8–15.1)	13.5 (12.2–14.6)	13.8 (12.7–14.9)	13.2 (11.0–15.0)
**Serum creatinine (mg/dL), median (IQR)**	0.90 (0.7–1.1)	0.9 (0.7–1.0)	1.0 (0.8–1.3)	0.9 (0.7–1.1)	1.2 (0.9–1.6)
**C-reactive protein (mg/dL), median (IQR)**	4.18 (1.2–9.2)	2.2 (0.7–4.8)	8.1 (3.5–13.9)	3.1 (0.9–8.4)	8.0 (4.8–16.0)
**Lactate dehydrogenase (IU/L), median (IQR)**	296 (229–367)	269 (210–330)	328 (264–419)	287 (222–346)	347 (275–527)
**International normalized ratio, median (IQR)**	0.96 (0.90–1.02)	0.95 (0.90–1.00)	0.98 (0.92–1.05)	0.95 (0.90–1.01)	1.01 (0.94–1.14)
**D-dimer (µg/L), median (IQR)**	593 (368–1344)	499 (307–940)	824 (467–1918)	550 (354–1083)	1161 (641–2834)

Abbreviations: IQR: interquartile range.

## Data Availability

The data presented in this study are available on request from the corresponding author. The data are not publicly available due to privacy concerns.

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
