# Peer review of "Risk of Seven-Day Worsening and Death: A New Clinically Derived COVID-19 Score"

_viruses, 2022, doi:10.3390/v14030642_

Round 1
Reviewer 1 Report
The article provided a feasible and easily-applicalble score to distinguish between SARS-CoV-2 infected patients who will encounter clinical deterioration or fatal outcome and those who will survive. There are several major/minor points to be addressed.
- It is suggested to present COVID-19 treatment strategies in these two patient cohorts in Table 1 and Table 2. Although the strategies may be mainly supportive, they underwent profound changes during the period as discussed in the article.
- The ROC curve of other cut-off values and different combinations of scoring factors should be presented.
- Figures could be beautified.
Reviewer 2 Report
Authors have used the retrospective data for this observational study to generate risk score for associating outcomes as either worsening or death from COVID-19 associated symptoms. Although both derivation and validation cohorts are identical except for different times of hospitalization, the design and analysis is very comprehensive given the limitations of matching for all the variants in the model. Overall, the methodology and the results are well presented. Few minor suggestions to improve the manuscript further.
- Is it possible to look at the association of the oxygen requirements based on the WHO ordinal scales with death outcomes? will there be any significant difference of those falling to score 1 or 2?
- Also please explain what does score 1 and 2 represents exactly for below the Table 2?
- Also create a separate section in the results section to highlight what factors really driving the severe outcomes like death from the multivariate analysis?
- Not sure if survival analysis was conducted apart from multivariate analysis
- During the multivariate analysis it is not clear if any covariates were included/controlled for?
Reviewer 3 Report
The authors proposed a new clinical, easy and reliable score to predict outcome in SARS-CoV2 hospitalized patients. This study is novel and the results will be useful for clinical practice.
Major concerns
- Did the authors considered serious underlying diseases of COVID-19 patients.
- The authors should use Multiple Regression,Machine Learning et al. to improve the manuscript.
Round 2
Reviewer 3 Report
Accept in present form.